# Examining Adolescent Tennis Participation in Contemporary China Using an Ecological Framework

**DOI:** 10.3390/ijerph19105989

**Published:** 2022-05-14

**Authors:** Longxi Li, Oliver J. C. Rick, Elizabeth M. Mullin, Michelle E. Moosbrugger

**Affiliations:** 1Department of Physical Education and Health Education, Springfield College, Springfield, MA 01109, USA; lli@springfieldcollege.edu; 2Department of Sport Management and Recreation, Springfield College, Springfield, MA 01109, USA; orick@springfieldcollege.edu; 3Department of Exercise Science and Athletic Training, Springfield College, Springfield, MA 01109, USA; emullin@springfieldcollege.edu

**Keywords:** adolescents, physical activity, ecological framework, thematic analysis

## Abstract

Physical activity and sport participation behaviors in children and adolescents are consistently shaped by surrounding ecological systems. Accumulating evidence highlights individual, family, peer, school and teacher, and macroenvironment elements such as policies that affect unstructured physical activity choices in youth populations. However, the reason for participation has not been fully interpreted from the perspective of the youth themselves, especially those from an Asian cultural background. In our study, we aimed to better understand the self-identified reasons for adolescents’ participation in non-organized or spontaneous tennis practice in contemporary China. Twenty-six adolescents and informants were recruited in mainland China and participated in semi-structured interviews to provide thick descriptions of their continued tennis participation behaviors. Data were coded and analyzed via NVivo 12. Four themes emerged: (a) Individual characteristics and self-interpretations of tennis culture; (b) microsystems mediating adolescents’ tennis participation; (c) barriers and obstacles impacting tennis participation; and (d) policies and macroenvironments. Adolescent tennis participation is a result of the integration effect of the sociocultural and ecological factors dominated by multifaceted ecological systems. As a particular vision of their physical activity experiences, adolescents’ interpretation of tennis and their broader worldview has been continuously reshaped by concurrent sport and educational policies.

## 1. Introduction

Physical activity (PA) and sport participation rates are on the rise globally [1]. However, less than 30% of children and adolescents worldwide meet the recommended PA guidelines of the World Health Organization, which was identified as a significant risk factor for global mortality [1]. The evidence is mounting that PA participation positively impacts physical and mental wellbeing, especially during adolescent development [2,3,4]. Therefore, motivating more children and adolescents to participate in PA practices that facilitate physical and social interaction should be given adequate attention, especially in the post-COVID-19 era. The three dominant categories for PA participation are physical fitness, team sports, and outdoor activities [1]. Of these three categories, fitness and outdoor activities have seen a more dramatic increase in participants, while team sports figures have risen to a lesser extent [1]. Tennis has been traditionally stable in participation internationally [5]. Although tennis is currently experiencing lowered participation rates in the US, there is a growing trend in Asian countries, such as China and India [6,7].

According to the International Tennis Federation’s report [8], Asia has more than 33 million tennis players, contributing 37.9% of the world’s tennis population (22.5% were contributed by China). Most adolescent tennis participation is associated with recreational practice and non-organizational patterns rather than professional and organized participation [8]. If these adolescent participants were counted, China would account for a considerable proportion of the global participation numbers. After the 2008 Beijing Olympics, a string of high-profile successes on the court (including Li Na winning two grand slam titles and Zheng Jie and Yan Zi winning the Wimbledon Championships doubles and the Australian open doubles titles) helped raise the profile of tennis in China. As Sun [9] highlighted, “Li Na and tennis has become a new business card for Chinese sports and even China” (p. 113). These progresses and glories engaged the Chinese public more broadly, and gradually knowledge of tennis increased. Furthermore, in the 2008 Beijing Olympics run-up, changes to policies made by the central party signaled a broader shift in sports development in China [10,11]. A core number of “global sports”, including tennis, have become the focus of Chinese state policy and investment since 2008 [12]. Although soccer, golf, marathon running, ultra-sports, and others have played a similar role, tennis is central to this contemporary global sport model [13,14]. For a generation of Chinese teens and people in their twenties that outnumbers the current population of the United States, questions considering how people, especially younger generations, commence participation in tennis become more complicated and intriguing.

Meanwhile, tennis is a practical and effective approach to promote adolescent physical fitness [15,16], cognitive and executive function [17,18], psychological wellbeing [19,20], and lifelong development [21]. In this way, the question of how to encourage adolescent participation in PA and sports, such as tennis, is emerging globally. However, little research exists regarding how this process plays out in national cultural contexts throughout Asia, such as in China and India, and in a specific global sport, such as tennis. Thus, the question must be answered by examining the nature of adolescent participation experiences in the sport of tennis and understanding how ecological factors function in their participation.

Physical activity, play, and sports involvement are potential vigorous to medium socializing agents, which are part of a life-long socialization process in an individual’s particular ecological context [22]. Bronfenbrenner [23,24] developed ecological theory, one of the few comprehensive frameworks for understanding sociocultural influences on development, to investigate the relationship between an individual and the environment. Gabbard [22] discussed that the significant point of the ecological theory is the interactive and cumulative effect on change regarding how the environment shapes an individual’s development and sports participation behaviors. As individuals vary their ecological and social environments, they learn to adapt to new situations. Bronfenbrenner’s [23] ecological theory placed the individual at a theoretical center surrounded by four environmental systems: the microsystem, mesosystem, exosystem, and macrosystem. Later, the chronosystem was added to interpret environmental events’ patterning and transitions over the life course and sociohistorical conditions [22,24]. For several decades, ecological models have been extended and developed in health promotion literature as the fundamental basis in the field of PA and lifelong development [25,26].

In addition, through the lens of the ecological theory, new evidence was gained that children and adolescents’ non-organized PA participation was significantly correlated with multiple layers of ecological factors [27]. More specifically, macro- and chronosystems factors including weather, national-level economy, public safety, and sport policy were identified as the most influential factors; individual characteristics appeared slightly more influential than microsystem factors on adolescent PA participation and engagement [27]. Hence, PA is a complex and multi-dimensional behavior determined by numerous biological, psychological, sociocultural, and environmental factors. Therefore, ecological theory was utilized as the theoretical framework in this study (see Figure 1).

In this study, we aimed to better understand the underlying reason for adolescent participation in non-organized or spontaneous tennis practice in contemporary China and to contribute to the extension of the ecological framework regarding relevance in societies outside of Western society. The study was guided by the following questions:What is the primary motive for adolescents frequently participating in tennis in mainland China?What factors or elements are associated with continuous tennis participation regarding the ecological framework in mainland China?How do adolescents in mainland China construct their worldview of tennis participation, a particular vision of their PA experiences? Moreover, how does that worldview shape their behavior?

## 2. Methods

Semi-structured interviews were utilized to explore the experiences of adolescents participating in tennis in China. Interviews were conducted when approval from the university Institutional Review Board (IRB) was granted. The study was also informed by a broad reading of sport and PA policy in China. This was not a systematic review of policy data; the policy was used to contextualize the analysis of participants’ responses.

### 2.1. Participants

Participants (7 female, 7 male) for this study were drawn from mainland China. Participants were purposefully delimited to adolescents who had actively participated in tennis for 3 to 10 years and were not affected by a particular disease, disorder, injury, or trauma at the time of the study (see Table 1). In addition, participants’ parents, friends, and teachers or coaches were considered informants (5 female, 7 male), and were recruited to help the primary researcher better understand participants’ experiences in tennis participation (see Table 2). One informant declined interview for personal reasons.

The first author determined the sample size, with sufficient information power regarding the aim of the study, sample specificity, use of established theory, quality of dialogue, and analysis strategy [29]. Interviews were conducted until the first author concluded the data had reached sufficient information power, or when no new information appeared in the interviews [29,30,31]. Participants completed the consent form and demographic information survey prior to the interview.

### 2.2. Instrumentation

Data were collected using a semi-structured interview guide and the assistance of the embodied function of Tencent Meeting for video and audio recordings. An interview guide had been developed based on the ecological frameworks delineated previously with age-appropriate questions for adolescent participants. Interview questions were open-ended and allowed the opportunity for probing and follow-up questions to occur (see Appendix A).

### 2.3. Procedures

Following approval from the IRB, the first author recruited and then contacted prospective participants. First, a maximum variation method of sampling from participants who visited tennis facilities in several study sites was used to establish a heterogeneous sample representative of the population with the broadest possible range of characteristics regarding the topic of interest [31]. The first author contacted tennis clubs in selected research sites to ask for their permission to post recruitment flyers. Flyers were posted near several highly visible tennis facilities in the area, such as tennis courts, training rooms, club lobbies, shops, and bulletin boards.

Furthermore, the snowball or networking method of sampling, a beneficial technique that can highlight the structure of the cohort being studied [32,33], was applied. More specifically, the first author identified potential informants based on the primary participants’ interview responses and contacted some informants directly to check their interest in participating in this study. In addition, considering tennis policy makers or administrators are a crucial component of the ecological system which is continuously shaping the macrosystem through policies and regulations, the first author contacted participants’ supervisors for permission to contact them to participate in this study.

Once participants were selected, the first author sent either the survey QR code or a link to their preferred contact address. Then, participants and their guardians were guided to complete the parental consent form and adolescent assent form; accordingly, a demographic form was completed by their guardians once the consent form had been signed. Online, synchronous interviews were conducted through the Tencent Meeting Web Communications software (Shenzhen, PRC) on a computer (Surface Pro 2; Microsoft, Redmond, WA, USA). After the interview, the audio recordings were transcribed verbatim in Chinese simplify; the translation process included a translating, checking, revise translating, and final translating procedure (see Appendix A).

### 2.4. Data Analysis

Thematic analysis of the collected data was inductive and comparative, beginning with the English version verbatim transcription of the interviews. Open, axial, and selective coding were used to generate initial codes, search for themes, review themes, and then to classify higher-order subthemes and categories [31,34,35]. The first phase of data analysis included translation and open coding of the interviews. Initial interviewing notes, comments, and memos created by the first author were generated during the open coding phase. Along this line, the groupings of initial codes and notes were organized into similar codes and initial subthemes of information during the axial coding stage. Finally, selective coding occurred as the data became saturated into major themes that clearly represented the phenomenon investigated [35,36]. Selective coding was used by the first author and the peer debriefer (O.J.C.R) to determine overall themes and categories of the data that emerged from the open and axial coding stages [31,35,36]. The data analysis procedure met the requirements of the 15-point checklist of criteria for thematic analysis [34]. All data were analyzed via NVivo 12© (NVivo12, QSR International, Melbourne, Australia).

### 2.5. Trustworthiness

Trustworthiness of the data was established through the following procedures: bracketing interview (confirmability), inquiry audit (dependability), member checking (credibility), peer debriefing (credibility), thick description (transferability), audit trail (confirmability), and ensuring results were derived from data (confirmability). The positionality statement was addressed prior to data collection and cited in Appendix A.

## 3. Results and Discussion

In this study, we aimed to better understand the underlying reason for adolescent participation in contemporary China. Data were collected from semi-structured interviews with adolescent participants (*n* = 14) and informants (*n* = 12) as evidence to answer the pre-established research questions. The following themes emerged from the interviews: (a) individual characteristics and self-interpretations of tennis culture; (b) the microenvironment mediating adolescents’ tennis participation; (c) barriers and obstacles impacting tennis participation; and (d) the macroenvironment and policies, the most prominent theme. The thematic framework of merged themes and research questions is illustrated in Figure 2.

### 3.1. Individual Characteristics and Self-Interpretations of Tennis Culture

The individual was positioned at the center of the concentric circles in Bronfenbrenner’s [23,24] ecological framework. The theme of individual characteristics and self-interpretations of tennis culture emerged from individual-related ecological factors and Chinese adolescent tennis participators’ (CATPs) worldview of tennis participation. In this study, CATPs’ continued tennis participation behaviors were influenced by psychological factors such as positive feelings, socializing, self-improvements, etc. Regarding CATPs’ individual interests, tennis was internalized with individual characteristics as a lifestyle, and facilitated participation behaviors and interpretations.

#### 3.1.1. Psychological Factors

The subtheme emerged from factors which were characteristics or facets that influenced CATPs psychologically and/or socially to engage in long-term tennis participation. Psychological factors were the most-reported reasons for tennis participations in CATPs, including several core categories: (a) simply like tennis; (b) positive feelings; (c) self-improvement; (d) socializing; (e) accountability; and (f) occupational objectives.

Most CATPs (*n* = 9) indicated the primary motive for tennis participation was “simply like tennis”, which involved being physically active through participating in tennis games. CATPs indicated that participating in tennis brought positive feelings, including a sense of accomplishment, feelings of uniqueness, reduced loneliness, and a sense of relaxation, which reinforced the CATPs’ interests in tennis. Playing in tennis games was “the opportunity to show my hard work” (Danielle) and could also bring “joy of winning” (Amy). Some CATPs referred to their appreciation regarding the opportunity to play tennis as the motivation for their participation. CATPs (*n* = 7) also qualified tennis as an effective communication conduit. Kevin’s mother indicated, “because of his [Kevin] tennis experiences, he [Kevin] has brought some friends around to play tennis…he [Kevin] is the central of his social group now”, which is one of the major reasons Kevin became interested in tennis. The majority of CATPs indicated that tennis connected to a sense of relaxation. The relaxation could be defined as the relief of a negative mood; as Max mentioned, “when I feel upset, I play tennis…that my mood suddenly improved after I played. Since then, every time I feel blue, I go to tennis court.” In addition, tennis was reported as an approach to cure loneliness (Ryan). CATPs believed that engaging in tennis could also balance the academic pressure at school; the balance related to positive feelings.

Tennis provided a space for CATPs to communicate with themselves. Danielle indicated that the self-communication ability was the most valuable outcome gained from her tennis experiences. The method of self-communication that CATPs learned playing tennis, “let me [Danielle] know myself better…feel energetic on onwards games and practices” and had been transformed and beneficial to her later tennis participation. During either a competition or regular play, tennis fostered a space for CATPs to conduct intrapersonal communications, also known as self-talk, which was widely used for enhancing sport performance and serving instructional and motivational functions [37]. In the current study, self-communications were found to be instructional functions in match competitions and motivate CATPs’ continued tennis participation. In summary, through tennis participation, CATPs were becoming more confident and determined from improved tennis skills and the improved ability to self-communicate. Self-improvements were also found in CATPs when tennis strengthened “capacity of concentration” (Amy) and enhanced “mental toughness” (Cloris), which further reinforced the continued participation behaviors in the long-term perspective. CATPs persisted in tennis because their thoughts had been valued and they felt responsible for their own decisions. The accountability was built on support and effective communications among parents, coaches, and CATPs. In addition, playing tennis was helpful for accomplishing CATPs’ occupational objectives. Max and Warrd mentioned that participating in tennis was parallel with their life goal, which is “running a tennis club in the future…help others better understand and participate in tennis”.

These results are consistent with previous findings in the children and adolescent PA participation literature [22,25,27,38]. According to Kemp and colleagues [38], behavioral and psychological traits made up the individual characteristics of adolescents and predicted their PA behavior. Furthermore, positive feelings and self-improvements would plausibly increase CATPs’ PA enjoyment and therefore tennis participation [39]. Li and Moosbrugger [27] discussed how greater PA self-efficacy may result in more moderate to vigorous PA, especially in boys. In the current study, diverse motives increased tennis self-efficacy and long-term goal setting, with future education- and occupation-reinforced continued participation, which broadened the previous findings. However, physical and behavioral factors found within previous research such as sex, pubertal development, and linguistic diversity [27,38,40] were not evident in the current study. In general, tennis games and practices provided opportunities for adolescents to experience success as well as failure and internalize varying experiences to their individual characteristics, which is pivotal to the development of independence and problem-solving skills in adolescents [22]. Accordingly, internalization of social and PA culture values became a lifestyle of CATPs, as both a process and a result of their tennis participation.

#### 3.1.2. A Lifestyle: Self-Interpretations of Tennis Culture

CATPs’ persistence in tennis was also attributed to several traits gained through long-term participation. These posteriori traits were grounded in and tightly related to CATPs’ tennis experiences, and could be reinforced in a supportive environment. Long-time engagement in tennis formed a pattern which kept motivating their behaviors and reshaping their interpretations of tennis. After a period of participation, some CATPs (*n* = 4) indicated their interest in tennis had been gradually shifted to hobbies and beyond. Max emphasized, “tennis became a very important part of my life, like my half-life”. Playing tennis was identified as a healthy lifestyle because CATPs believed tennis is a lifelong sport and could help them to become healthier. Long-term engagement in tennis is a process of integrating positives from tennis participation into a new style of life; it is not simply repetitive behaviors. CATPs’ interpretations of tennis and domestic tennis culture interacted and tightly correlated with participation behaviors. Emerged categories included elegant, niche, high-end, and internationalization, which demonstrated the core of CATPs’ interpretations of tennis and its culture. Tennis was acknowledged as an elegant sport which was more “civilized”, with less physical contact compared to invasion ball games. Rules and conventions in tennis were identified by CATPs and informants as a “moral code” to regulate players to be gentle and demonstrate fair play in daily practices as well as competitions. Good manners and dressing etiquettes in tennis were especially valued by CATPs and informants.

Culture and broad social trends lie behind the manifest appearances of an everyday life [41]. From a sporting sociological imagination, Sage [42] addressed, “any adequate account of sport must be rooted in an understanding of its location within society…the essence of sport is to be found within the nature of its relationship to the broader stream of societal forces of which it is a part” (p. 14). Tennis participations are sphericity social events in ecological systems which reflect the mechanism of the society. Within the event that composes physical culture, individuals craft and negotiate their embodied subjectivities and identities through multi-layered relations of power in contextually contingent ways [43]. Given that the influence of physical culture was largely unexplored in the field of youth PA and health promotion, one of the major advantages of this study was providing a first-person perspective with a lens of culture through an ethnographic approach [31,44].

Tennis culture has never had a fixed definition, instead it’s meaning is constantly being constructed by those participating in the sport, inflecting it with their particular class sensibilities and expressing their dispositions generated from a certain cultural background. Several interpretations of tennis culture, such as elegant, niche, and the high-end international platform, reflected the merits of tennis and aspirations from CATPs’ perspective. Social groups exhibit different motivations for involvement in PA, and the members of a specific class grouping tend to be inclined toward specific PA [45]. In our view, physical culture is a product and producer of tennis, and the self-interpretation of tennis culture became one of the interconnections between sports and the social class identity. Being represented as a complex cultural phenomenon, tennis had been re-interpreted by CATPs, and ultimately, participating in tennis became a lifestyle of CATPs, which unprecedently secured their persistence in tennis participation.

### 3.2. Microenvironment Mediating Adolescent Tennis Participation

The microenvironment represented the combination effect of the immediate contexts in which the individual partakes in adolescent tennis participation behaviors. Several social agents (e.g., family, peers, and coaches) within the microsystem were found with critical roles in reinforcing and mediating CATPs’ continued tennis participations.

#### 3.2.1. Family

All CATPs acknowledged that parents firstly introduced tennis to them and were their primary support for tennis participation. Some CATPs (*n* = 2) learned and played with their parents, and some CATPs (*n* = 5) were influenced by their parents’ current or previous PA patterns. Parental social networks or occupations (e.g., working in sports-related jobs or having relations who had coached tennis) provided privileges for their children regarding easier access to coaching resources and tennis facilities, and even the expense of practices. Therefore, parental attitude towards sports and tennis was identified by CATPs (*n* = 12) as a dominant factor in tennis participation. The popularity of tennis was unprecedentedly high in China “when Chinese player Li Na won the Grand Slam in 2011” (Joe). As tennis gained popularity, more Chinese parents created a positive family environment and encouraged their children to play tennis. Parents’ life experiences and parenting philosophy were blended in the decision-making process of tennis participation. Hence, parents’ considerations might go beyond tennis as a sport, e.g., focusing on the benefits for their children’s health and lifelong development or simply respecting and supporting their children’s choices. Warrd’s mother argued that she tried her best “to improve Warrd’s understanding of sports and education…health first and exam scores second”.

Moreover, parental logistic support, including creating opportunities for children to play, providing transportation, paying expenses, and lasting effective two-way communication, was equally important to CATPs’ tennis participation. CATPs’ parents shared a similarity; they believed children’s interests were more important than anyone else’s. Ryan’s father said, “I am glad tennis is his [Ryan’s] own decision and he can follow his ‘inner voice’.” Effective parent–children communications (parents as a careful observer and listener) were found beneficial to increase CATPs’ self-efficacy in tennis and general PA. Higher self-efficacy resulted in more PA participation in adolescents [46,47], and so, CATPs could be more determined to be consistent in tennis participation. Besides, participating in tennis occurred under certain conditions, especially relating to the expenses. Max and Ryan were concerned about high expenses, and they were grateful to their parents for their financial support for court rentals, equipment, coaching, and transportation expenses.

Family relations are the “primary sociocultural influences on sports involvement and PA during childhood and adolescence” [22] (p. 376). As the core of family, parents were directly responsible for the communication of cultural content to the growing child. Appropriate parenting skills and health behaviors would positively construct childhood PA and were vital in cultivating an autonomous environment with flexibility and respect for the CATPs’ decisions [48], which promoted unstructured PA behaviors [49]. In addition, parental attitudes toward a healthy lifestyle and sports and their logistic support were important in formulating children’s unstructured active play across genders [27,50]. Parents’ PA patterns, occupations, and social networking determined parents’ attitudes and beliefs towards tennis. Children of more educated parents and higher family socioeconomic status appeared to be more physically active than their counterparts [38,40]. In this study, commonalities can be found across families; more than 83.3% of parents had higher education background, worked in full-time positions, and reported middle or higher socio-economic status (see Table 2). The high profile and secured family environment reinforced CATPs’ frequent and continuous tennis participation.

#### 3.2.2. Peers

Peer influences were found in two main aspects: shared interests and mutual support. Within the cohort, peers were able to exchange positive feedback and support each other’s development, which significantly motivated CATPs in tennis participation.


*“Peers’ influences are quite big to him [Kevin]…friends around him create a positive and inclusive atmosphere when they participate in tennis. The desire of competitive was turned on…especially boys in this age…those positive interactions has further motivated and boosted his enthusiasm for tennis.” *
(Kevin’s mother)

Adolescents with higher perceived social support from friends and social networks demonstrated higher levels of PA participation [51,52]. In the current study, positive impacts from peers such as similar interests and goals and supported CATPs in game competitions as well as daily participation. Undoubtedly, peers were a promoting factor which motivated CATPs’ tennis participation behaviors and enjoyment.

#### 3.2.3. Coaches

Coaches were found to be supportive to CATPs’ (*n* = 11) tennis participation, mainly for psychomotor skill development and mental support. “Mensch” was frequently used to represent a coach by CATPs and informants. Max trusted and looked up to his coach very much and kept his coach’s words in mind. Four others indicated they were grateful to their coach, who influenced them the most as they grew. Gabbard [22] indicated that coaches have the potential for being an influential agent toward adolescents’ PA. In this study, coaches had a profound influence on CATPs’ skill development and personal growth beyond tennis. Through the coaching–participating process, trust and reverence were built on an equal footing between CATPs and coaches. As an ancient Chinese proverb described, *yi ri wei shi zhong shen wei fu* (a teacher for a day is a father for a lifetime). Especially in a Chinese cultural context, coaches would promote CATPs’ continued tennis participation and played an irreplaceable role in enhancing self-concept development [53].

In addition, ecological systems should not be interpreted independently [22,24]. The microsystem is interconnected with the extended or inter-family factors and beyond through the mesosystem [54]. In the scope of the current study, interactions between two or more microsystems, such as the relationship of family to school experiences or of family to neighborhood, were associated with CATPs’ tennis participation and also underlined the importance of the mesosystem on adolescents’ PA.

### 3.3. Barriers and Obstacles Impacting Tennis Participation

CATPs experienced varying internal and external impediments during tennis participation. Internal barriers and obstacles were identified as personal or psychological factors that hindered tennis participation. Skill-specified obstacles and injuries were mostly referred to by participants. Meanwhile, external barriers and obstacles were factors beyond the intrapersonal level which negatively influenced tennis participation, including teachers’ resistances, heavy schoolwork, social norms and body image, insufficient tennis resources, expenses, and, most recently, COVID-19 restrictions.

#### 3.3.1. Skill-Specified Obstacles and Injuries

CATPs (*n* = 6) identified that losing in a competition discouraged their passions and motivations for tennis. Unproficiency in skill performance, which CATPs commonly referred to as “bottleneck”, was reflected by CATPs as the major reason for losing games. This lowered the enjoyment of participation and further impaired CATPs’ enthusiasms in tennis. Along this line of consideration, experienced coaches or physical educators might be considered as a pivotal external assistance on CATPs’ motor skills and development [22]. Additionally, improving coaching standards was a priority so that the youth could be led by individuals who had been trained in sport skills, tactics, and safety protocols [55]. Addressing the lacked role of coaches in schools or communities might help to engage and retain CATPs in both organized and non-organized tennis participation. Meanwhile, injuries, caused by undeveloped physical skills and cognition, were one of the main reasons for inconsistency in habitual sport participation [56]. Although recreational tennis participation had a relatively lower intensity and was less competitive compared to similar-aged professional athletics, injuries were commonly acknowledged as internal obstacles by CATPs. Given the influence of injuries and experience in non-organized youth tennis was not fully discussed, future research is encouraged on exploring effective prevention and rehabilitation strategies for this group.

#### 3.3.2. Schoolwork and Resistance from Teachers

Teachers in school were recognized by CATPs (*n* = 8) and informants (*n* = 10) as one of the crucial barriers on CATPs’ tennis participation. Student academic performance, especially grades and school enrollment rate, was one of the key performance indicators for teachers (Louis). Schoolteachers commonly expressed negative attitudes regarding tennis participation because they believed participating in tennis and other PA would disturb students’ time allocation for schoolwork. Amy mentioned that her schoolteachers “suggest me stop playing tennis when exam day is approaching…they [teachers] worry playing tennis or involving PA might swallow my study time and disturb my mindset [in the exam]”. Stereotypes of PA and sport participation were found deeply rooted in Chinese middle schoolteachers.

Indeed, the concern was not groundless, as K-12 students were experiencing high-intensity academic pressures in and after school (e.g., working on considerable homework and exams), especially in middle-high school. Jerry noticed “there are not as many tennis players in the 12–18 age groups as the 6–12 age groups, it drops dramatically”. Notably, a high-intensity curriculum and considerable amount of schoolwork resulted in enormous pressures, possibly caused by the examination-oriented education system [57,58]. As a result, tennis and school became a multiple-choice question for CATPs who could “only chose one at the time, it’s non-negotiable!”, as Danielle grumbled. Facing the dilemma, which is a “common phenomenon” according to Kevin’s mother, CATPs and parents evaluated the situation carefully and made tough decisions. Tennis only reflected a piece of the ‘iceberg’ of the status quo of youth PA participation in mainland China. Adolescents were not only struggling in participating in tennis, but also had insufficient time to engage in regular PA or exercise in general. The average PA duration among Chinese school-aged children was low in national normative data, and only 30% meet the recommendation, especially evidenced among junior middle and junior high school age-groups [59]. Consequently, a massive number of teenagers gave up tennis following enrollment in high school.

Moreover, researchers argued that youth PA and sport participation was not taken seriously in China [60]. There might be several reasons, such as the marginal position of school physical education (PE) curriculums and imbalanced public resources between elite sport and mass sport [60]; schoolteachers’ biases on sport and PA [61]; or the unprecedented school enrollment pressures forcing adolescents away from regular PA and exercise [62]. A new governmental policy was established aiming to advance the integration between sport and education (*ti jiao rong he*) in 2021 [63]. The policy highlighted that healthy body and academic performance should be treated as equally as important and would be evaluated together in students’ gradebooks. Although it is too early to evaluate the effectiveness of this policy, students’ PA participation and health education have seemingly raised in status from the national level concurrently. However, the school and tennis ‘confliction’ was still undermining CATPs’ primary motives concerning tennis until now.

#### 3.3.3. Social Norm and Body Image

Adolescents mature socially and culturally just as much as they do physically, and “they learn and are channeled into gender specific physical cultures, bodies, and identities” [45] (p. 6). CATPs mostly participated in tennis in outdoor environments; due to sunniness, they easily tanned. However, tanned skin was not aligned with the mainstream social norm regarding body image for girls and women in modern Chinese culture. Cloris’s experience uncovered female CATPs’ concerns and how the guidance from parents protects CATPs’ tennis participation in China.


*“She [Cloris] noticed that playing tennis making her skin tan, she was very concerned about it…because it [tan skin] is the opposite of the mainstream of beauty-appreciation in China. I tried to explain to her and help her not bothering from it. I persuaded that people in Europe are white, but they all prefer to spend considerable money on tanning or sunbathing. It [tan skin] is fashion! Whiteness is not the only standard of beauty…Even if you are tanned [due to tennis], after a while, you will back to the normal…In such a big social environment, it is difficult for me to change all her thoughts. I told her [Cloris] a truly beauty has a healthy and fit body rather than simply pursuing skin color and slim body shape…she [Cloris] slowly accepted it, and sometimes she looks herself in the mirror after the conversation…rambling, ‘my skin tone and legs are not looks that bad actually’.” *
(Cloris’s father)

Cloris’s father emphasized that “beauty is a spectrum which should be diversified in nature, however, the beauty-appreciation in China was getting more and more abnormal which is misleading to children”. Evidently, women are influenced by pervasive cultural norms related to the way that feminine bodies should act and look. Social norms and body image in China were distinctively aligned with the Western mainstream physical culture, “hegemonic gender norms within Chinese culture have created challenges for athletes, markets, and national politics” [64] (p. 524). The ‘male superiority’ and ‘inability constitutes the very virtue of a woman’ are rooted in concepts of Confucian [64]. In Butler’s [65] perspective, gender is culturally formed, and it is most important to resist the violence that is imposed by ideal gender norms. In the last decade, successes and glories in Chinese women’s tennis challenged the *traditional* social norm and physical culture, and at same time promoted China on the global stage [10]. Further, the public appeared to gain higher expectations of female athletes and women’s sports teams.

Other parents believed tennis was perfect for adolescent girls because tennis was seemingly safer and more elegant than most other sports.


*“I [Cindy’s father] have been playing soccer since I was a child and I end up my professional career with innumerable injuries. I [Cindy’s father] don’t want Cindy to do the same…alternatively, maybe participate in sports without physical contact. This kind of fierce physical confrontation is more suitable for boys. Don’t give me wrong, tennis is suitable for both boys and girls…just that tennis is more beneficial to girls’ body shaping…be more confident, gentle and graceful…” *
(Cindy’s father)

Aggressive behaviors and physical contact are commonly seen in sports competition; however, they are not respected as traditional Chinese gender expectations for girls [64]. Compared to sports with intense physical competition, parents and CATPs’ interests in tennis or other etiquette sports were still reflecting the remaining heritage of traditional expectations on women. Furthermore, barriers impacting PA participation in adolescent girls were also caused by PE curriculums in school which emphasized muscularity [27]. Although concerns around appearances were linked to the ‘socially accepted’ feminine body shape or norms of female sport, we believe CATPs’ interpretations of the body image and physical culture were able to be influenced and reconstructed with parental guidance and support.

#### 3.3.4. Tennis Expenses

Family disposable income spent on CATPs’ hobbies was found to be related to tennis participation. As court renting and coaching fees were increasing dramatically, it became a considerable expense for a family. In addition to court renting and coaching fees, transportation and summer camp expenses would be added to the overall budget if CATPs qualified to participate in regional or national games. According to Cloris’s father, “the estimated annual expense for tennis was 200,000 yuan [RMB] which is close to middle class family annual income in China”. A majority of parents acknowledged finance as a stressor, while coaching fees and tournament traveling expenses were the core [66]. High expenses limited and even stopped CATPs’ tennis participation in China. Unlike costly sports such as tennis and golf, basketball and badminton were more common in China because they were more accessible and affordable.

#### 3.3.5. Shortage of Resources

Limited tennis facilities in schools and neighborhoods were addressed by CATPs (*n* = 9) and informants (*n* = 4) as an obstruction, especially CATPs from the north regions and small cities. Public tennis courts were mainly built in the city sport center and could not fulfill the demand of the increasing tennis population. A shortage of sport facilities resulted in children’s and adolescents’ physical inactivity [67]. Instead, a safe, well-equipped, and supportive school and neighborhood environment would significantly increase weekday and weekend MVPA in children and adolescents [27]. The insufficient amount of tennis facilities was destroying CATPs’ enthusiasm in tennis, and meanwhile, increasing the tennis expenses to some extent. Lastly, confronting the global pandemic, CATPs (*n* = 5) indicated their tennis participation had been interrupted. New obstacles emerged due to city lock down and associated restrictions, which dramatically increased sedentary behaviors and screen time, and at the same time, decreased PA participation in children and adolescents and further aggravated the shortage of sports resources [68,69]. Although restrictions to the public are being removed gradually, the long-term influences were still largely uncovered on how COVID-19 changed patterns and perceptions of sports participation in adolescents.

### 3.4. Macroenvironment and Policies: The Most Influential

Switching the angle from insulated parts to the larger whole, the macroenvironment factors were shaping CATPs’ judgement and worldview on tennis participation. The macroenvironment was equivalent to the macrosystem, the outer edge of the ecological systems. The macrosystem involves “the culture in which the individual exists” including PA patterns or behaviors, beliefs, and culture products which were passed for generations [22] (p. 24). Macrosystem factors were the highest correlational factors among the ecological system on children and adolescents’ non-organized PA [27]. Hence, the macrosystem seemed an ‘indirect’ but dominant factor in CATPs’ tennis participations. The emerged subthemes include tennis role models and events, concurrent college admission policies, and geographical locations.

#### 3.4.1. Role Models

Eight CATPs emphasized that tennis role models, especially Chinese female tennis player Li Na, fired up CATPs’ enthusiasm for participating in tennis. Cindy was greatly excited when she was asked about her tennis role models.


*“I think tennis culture in China is Li Na, vice visa. She [Li Na] is an icon…Although I might not have a chance to be a tennis pro like her, Li Na is still my super star. Li Na’s personal experience is not roses all the way, her [Li Na’s] story cheers me up when I feel down.” *
(Cindy)

Li Na’s success was ground-breaking and convinced youth to believe that “the possibility of winning the championship is not determined by your genes but your effort” (Cloris). In short, Li Na was generally reported by CATPs as a positive influencer who fueled their passion in tennis participations. As discussed previously, tennis successes in Summer Olympic Games and Grand Slams improved Chinese global image and produced numerous role models for the young generation. Role models were found as a promising factor to child and adolescent PA participation [70,71]. As catalytic, the *traditional* social norm and gender expectations on female athletes had been challenged and promoted by role models [64]. Lead by Li Na, tennis role models upscaled tennis and increased CATPs’ participation, especially in girls and women, in China.

#### 3.4.2. Tennis Events

CATPs (*n* = 13) indicated tennis events increased their interests in tennis, especially seeing and even interacting with successful tennis athletes. Max mentioned he “travels to Beijing every year for China Open…their [professional athletes’] performances and the authentic atmosphere making tennis even more attractive”. As more international tennis tournaments launched in mainland China (13 international tennis tournaments were hosted in China prior to COVID-19) attending tennis events became an approach for the youth population to know about tennis [72]. Attending tennis events could be a novel way for Chinese youth to be involved, gain in-depth understanding on tennis culture, and further shape their worldviews of tennis.

#### 3.4.3. Geographical Factors

Tennis was developed unevenly among cities and regions in mainland China. Southeastern coastal cities and provinces, such as Shanghai, Zhejiang, Guangdong, and Jiangsu, were recognized as the most developed regions in tennis in the study. Regional economic advancement tightly correlated with sport participation rate in mainland China [73]. The current research provided new evidence that tennis, as a high-expense middle class sport, amplified the impact of the imbalanced economic development and resources maldistribution on sport participation in mainland China. In addition, southern regions were more attractive to CATPs because of the ideal ambience. Key terms such as “pleasant climate”, “fresh air”, and “blue skies” were used in describing cities in the south whilst “cold and haze weather” was used for the north. Ambient temperature and rainfalls had substantial effects on PA, such as daily steps count, in prior research [74]. Particularly, air pollution became a serious concern in northern China, which was detrimental to the public health and decreased outdoor exercises [75,76]. As a largely outdoor sport, cold temperature and air pollution aggravated the imbalanced tennis participation between regions.

#### 3.4.4. The Policy ‘Shortcut’

CATPs’ primal interests in tennis were found either related or later shifted to future educational considerations. Parents believed that mastery in sport could equip children to be more competitive in their future development. As one of the options, tennis was identified by CATPs and parents as an attractive ‘shortcut’. Danielle, Rose, and Eason shared their perspectives that, “playing tennis means an earlier way of going to college and a better life”. Most CATPs (*n* = 12) acknowledged that playing tennis is beneficial to help them accomplishing future goals, such as enrolling into a prestigious university, which was one of the latent motives for CATPs. As long as skill level meets the standard, tennis skills could be identified as “sport specialty”, and the “bounce in points” would be awarded to applicants in college or even middle school entrance exams. Meanwhile, higher education institutions had also lowered the cut-off grade for applicants with athlete identities in admission, which attracted more adolescents to gain a tennis background in China. In addition, a group of CATPs and their parents chose tennis over other sports, surprisingly, as tennis expenses are relatively high. Parents explained that the high expense could plausibly deter the majority from tennis; the limitation became an advantage for a minor social group able to afford the expenses.

Nevertheless, the concurrent sport policy was not the single reason that made tennis a new favorite among Chinese adolescents. Facing the intensified educational competitions, academic performance could not make children stand out or prove ‘excellence’. Exams are no longer the only approach for Chinese youth to be elevated to the higher education nowadays (Louis). As a result of the domestic hyper-competitive education status quo, tennis became a gateway for CATPs to ‘escape’ the unprecedented fierce educational competition. Even though blending tennis and K-12 education together is similar to “gambling”, parents were still willing to take the risk and invest a considerable amount of time and money to bridge the gap for CATPs to higher education.

Overall, Bronfenbrenner’s [23] ecological model was examined under a Chinese social context in this study. Adolescent tennis participation in Contemporary China reflected the Chinese characterized socioecological environment as a producer and product of tennis culture and lifestyle. The physical culture of tennis fundamentally influenced parental perceptions and attitudes and then CATP’s tennis participation patterns. Ultimately, domestic educational environments, policies, and parents’ perceptions of tennis were found to interact and shape the CATPs’ lifestyle and worldview, which clearly crossed concentric ecological circles and linked them together.

## 4. Practical Applications

Practical applications of this research in the fields of PA, especially tennis participation and health promotion in adolescents, include (a) establishing and enhancing novel collaborative partnerships between students, schools, and coaches and (b) improving the pediatric health knowledge base of schoolteachers and parents.

In this study, as an obstacle rather than a supportive factor, school did not supply adequate resources for CATPs in developing their interests and skills in tennis. A practical collaboration between students, schools, and local tennis clubs could be established to foster a positive tennis atmosphere in schools. On the other hand, gained from interviews and current literature [60], parents and schoolteachers’ knowledge base on PA and healthy development could be improved. Although sport and educational policies were the root cause, we believed schoolteachers’ and parents’ perceptions towards tennis and sports could be promoted to counterbalance the consequences of incomplete polices and regulations. For instance, delivering workshops or presentations in schools and communities to schoolteachers and parents could be an effective route for them to learn and rethink the key role of PA and sport participation in children development. Especially during age of 12–18, the phase of growth and refinement, perhaps the “most significant motor behavior changes is seen at the time of puberty and the accompanying growth spurt” [22] (p. 14). Additionally, puberty is the one of the major landmarks in adolescence and human development. In this transitioning phrase, growth in height and weight are accelerated, and “the degree of logical and abstract thought increases as well as a concern about identity and independence” [22] (p. 12). Cloris is not the only CATP who is concerned about their appearance after years of outdoor tennis participations. It is promising that mass media and public in the mainland are gradually changing and refining their attitude toward women and women’s sports, which hopefully will bring significant changes to social norms and gender expectations in the future Chinese society [64]. Gaining a basic and scientific understanding of development would be beneficial for schoolteachers and caregivers to provide appropriate guidance to CATPs on their tennis participation as well as for the whole generation to grow healthier and be more physically active.

Lastly, limitations of the current study must also be addressed. The primary limitation of this study was the participant recruitment and selection process within the data collection. Participants or their guardians’ selection was limited to only those who saw the recruitment flyers through advertisements in local clubs and/or social media. This certainly was a limiting factor on recruitment and could have curtailed the diversity of respondents in the study to reflect the entire community of adolescent tennis participants. Another limitation of this study was the reliance on interviews as the only data collection technique. Due to the COVID-19 pandemic, global travels had been suspended. Thus, observation was not feasible. To control the limitation and minimize the bias, the researcher recruited sufficient participants and informants to reach the considerable level of information power [29,30]. Future researchers are encouraged to integrate ethnographic interview, policy analysis, and careful field observation to extend the current findings.

## 5. Conclusions 

This study provided a novel perspective to examine the primary motives and ecological factors for adolescents’ continued tennis participation in mainland China. Adolescents’ tennis participation patterns were not shaped by a simple factor; rather, effects of manifested ecological factors were observed on each individual and their tennis experiences. Individual characteristics and microsystem factors were found as effective mediators boosting CATPs’ continued tennis participation; macrosystem factors played a dominant role in cultivating a blended physical culture and beliefs around tennis in the Chinese society. As an entertaining PA, lifestyle, and gateway to the higher education, tennis participation was likewise a dual process of engaging in PA and internalizing the Western physical culture into the Chinese society. Playing tennis constructed CATPs’ interpretations of tennis culture and worldview; concomitantly, their worldview shapes the CATPs’ behavior. To our best knowledge, the current research was the first study which shed light on adolescents’ tennis participation in mainland China through an interview-based approach, and also contributed to extend the ecological framework regarding its relevance in societies outside of Western society.

As a follow-up to potential future directions reported herein, we recommend future explorations could be made via a concise and practical model which would be more focused on the individual. Especially, individual characteristics or intra-person factors were undervalued in previous literature but critically influenced adolescents’ PA participation and the life-long development. The Process Person Context Time (PPCT) model is considered as the primary choice which is more applicable than the general ecological framework, focusing on the individual and interaction between organism and environment [22,77,78]. Additionally, better knowledge of the mechanism of the internalization process and CATPs’ lifestyle on tennis participation would further support and promote unstructured PA and tennis participation in this age group. Based on the findings in the current study, we summarized that adolescents’ tennis participation was a result of the integration effect of the sociocultural and ecological factors which were dominated by multifaceted ecological systems. Ultimately, tennis is more than just a modern sport or type of PA to adolescents in Chinese society; rather, it is a spectrum of the cultural phenomenon.

## Figures and Tables

**Figure 1 ijerph-19-05989-f001:**
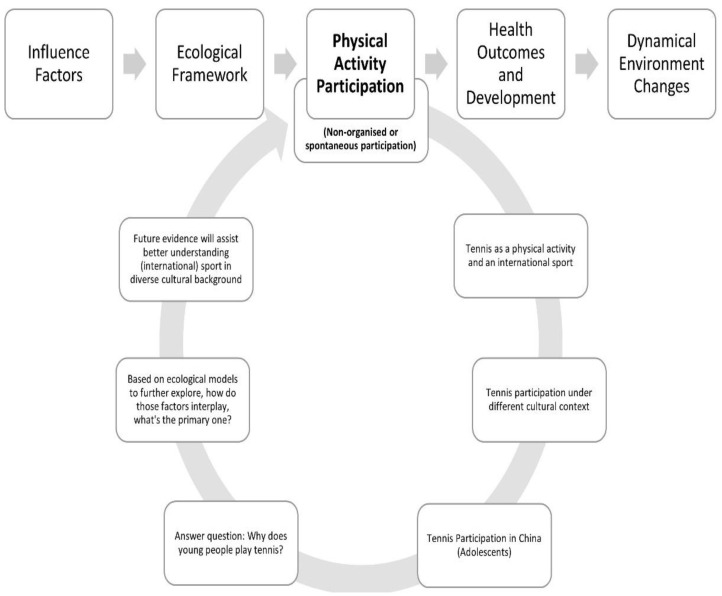
Diagram of the study design, aims, and initial questions.

**Figure 2 ijerph-19-05989-f002:**
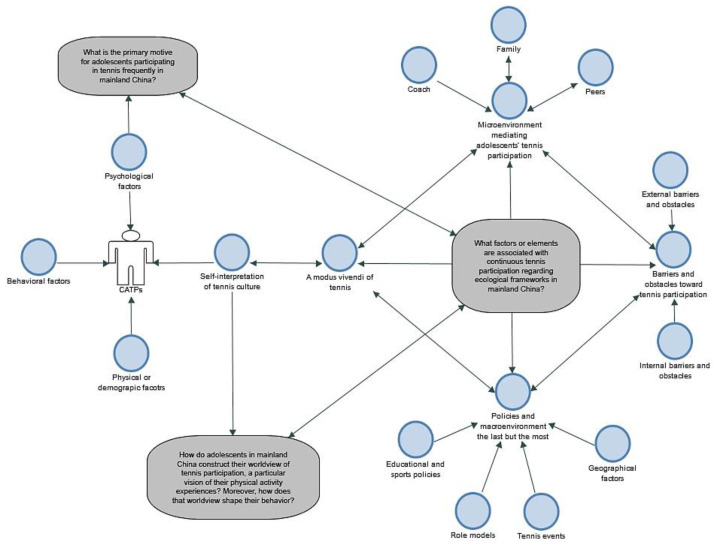
Thematic framework of merged themes and research questions.

**Table 1 ijerph-19-05989-t001:** Characteristics of participants (*n* = 14).

Pseudonym	Gender	City	Region	Age	Exp	Grade
Max	Male	SH	South	17	10	12
Warrd	Male	SJZ	North	16	10	11
Cindy	Female	SJZ	North	17	7	11
Ryan	Male	SJZ	North	17	3	11
Amy	Female	SJZ	North	14	5	8
Annie	Female	SJZ	North	12	3	5
Cloris	Female	SH	South	16	5	10
Danielle	Female	SJZ	North	15	6	9
Kevin	Male	BJ	North	13	6	6
Eason	Male	SJZ	North	15	6	9
Levi	Male	SZ	South	15	7	9
Jing	Female	SZ	South	14	7	8
Rose	Female	SZ	South	15	7	9
Hao	Male	SZ	South	15	8	9
*Mean* *SD*				15.07 1.44	6.43 2.03	9.07 1.94

South and north regions were identified geographically based on the Yangzi River [28]; Exp = Experience, which was self-reported using the question “how long have you participated in tennis?”; Sites refers to the city which the participants currently live in; Grade = Grade level of participants; SH = Shanghai; SJZ = Shijiazhuang; BJ = Beijing; SZ = Suzhou; SD = standard deviation; Means and SD were computed for age, experience (years), and grade level of participants.

**Table 2 ijerph-19-05989-t002:** Demographic Information of Informants (*n* = 12).

Pseudonym (Admin)	Informants				
	Role	Age	Educational Background	Occupations	Family SES
Cloris	Father	56	Bachelor	Self-employed	Upper
Max	Mother	55	Bachelor	Full-time	Middle
Warrd	Mother	45	Diploma	Full-time	Middle
Cindy	Father	46	Bachelor	Part-time	Middle
Ryan	Father	45	Bachelor	Full-time	Working
Amy	Mother	40	Diploma	Full-time	Middle
Annie	Mother	37	Bachelor	Full-time	Middle
Jing	Coach	36	Graduate	Full-time	Middle
Rose	Coach	36	Graduate	Full-time	Middle
Kevin	Mother	45	Bachelor	Full-time	Middle
Eason	Father	46	Graduate	Full-time	Lower
Levi	Coach	36	Graduate	Full-time	Middle
Hao	Coach	36	Graduate	Full-time	Middle
(Louis)	CTAM, RTO	44	Doctoral	Full-time	Middle
(Jerry)	CTAA, RTO, CTAM	35	Bachelor	Self-employed	Middle

SES = Social Economic Status; Diploma = high school degree or lower; Admin = administrative informant’s pseudonym name; CTAM = China Tennis Association Member; RTO = Registered Tennis Official; CTAA = City tennis association administrator.

## Data Availability

The datasets used and/or analyzed during the current study are available from the corresponding author upon reasonable request.

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
