# Peer review of "Examining Adolescent Tennis Participation in Contemporary China Using an Ecological Framework"

_ijerph, 2022, doi:10.3390/ijerph19105989_

Round 1

Reviewer 1 Report

This is a very well founded and executed investigation. Both the theoretical framework and the methodology are adequate. The results and the discussion are relevant and the conclusions provide new data for future research. In addition, it opens new lines of research.

Reviewer 2 Report

I find it an interesting study. The introduction is well founded. The methodology is clear. The results and discussion chapter provides good information. With the aim of making the reading of this part more pleasant, I would recommend the authors to reduce the information, since it is exhausting to read (it does not mean that it is bad).
Congratulations, it is a study that provides pioneering information in the study area.

Reviewer 3 Report

Dear Authors, your paper is very well written but seems to be incomplete in

  1. its conclusions;
  2. its supplementary material;
  3. explanation, use, and extension of the ecological framework.

The text is narrative, this is exceptionally well done, but has the disadvantage that the crisp details: what is the check with the currently known ecological framework and how it is extended, seems/is missing.

  1. Some sub results are concluded in the main text of your methods. These do not seem to be covered or copied in the conclusions. See below for the detailed remarks.
  2. I did not find in the three Supplements A, B, and C a report of the data analysis with the NVivo software (mentioned in Lines 21, 188). See some of my items below.
  3. An observation is that you promise to test a framework for your result. But this is absent in the main text. See some of my items below.

Details, typos, suggestions for improvements, and discussions

For your reading convenience, I sometimes propose to be inserted words in sentences in bold.

  1. Lines 2,3 You do not seem to use the Ecological Framework.

  2. Line 72, The Introduction is long. I propose to split it, by giving the subhead “1.1 Ecological factors” between lines 72 and 73.

  3. Line 97 – 98, Figure 1, I recommend putting the box at the right-hand top "Dynamical Environmental Changes" in the ellipse at the center.

  4. Line 102, where is the extension of the ecological framework, as planned here?

  5. Lines 106 – 107, here you promise too much. I do not find a comparison of your results to ecological frameworks. I do not find the comparison of your work with the ecological framework.

  6. Line 125 and further, the pseudonyms in Table 1 are reused in Table 2, except “Danielle” in Table 1. This is unclear. Please explain why the (admin) names are permuted, or is the parent name pseudonym for the name of the child?

  7. Line 130, the term “illustrated” seems wrong to me. Should it read “computed”?

  8. Between Lines 131 – 132, why is Danielle missing here? See my remark 6.

  9. Line 135 - 138, what is the numerical value of the power test for information sufficiency? Please put this in a statistical supplement.

  10. Line 170, PRC is mentioned here, while other cities are not located in the People’s Republic of China. Should PRC be mentioned here?

  11. Line 188, I suggest giving the details of the firm and the software in the references list. With such reference, it can be more complete. The NVivo is also a company and not just the software. In European universities is the NVivo encapsulated in workbenches. I guess it is better to express more details about the product (and its use).

  12. Line 205 – 206, the font in the boxes is too small for reading.

  13. Lines 233 - 234, 346, here you put subjects Kevin, Ryan between squared brackets: [Kevin]. Why different from other subjects? Line 231 has (Amy), and line 240 has (Ryan) in different brackets. Please make this consistent.

  14. Line 463, for the European readers, I propose to include “governmental” between “new” and “policy”.

  15. Line 468, What does mean “ . . from the national level concurrently”? What is “confliction”, is it a typo?

  16. Line 490, I think that “are misleading” should read here “is misleading”.

  17. Line 510, “… were not respected as traditional Chinese gender expectations …” is unclear to me. Could it read “ … were not perceived as traditional Chinese gender behavior … “?

  18. Lines 418 - 420, convey a conclusion.

  19. Lines 601 - 602, convey a conclusion.

  20. Line 676, concludes that you provide a novel perspective. But it seems not compared to the current view in the ecological framework. Could you explain where the novelty is checked?

  21. Line 689, you say that the ecological framework is extended. Please explain the difference with the previous version of the framework.

  22. Line 695, tells about the PPCT. This pops up here in the conclusions. This is incomplete, it should be explained before. Does this belong to the test of the ecological framework? And does it also belong to the Statistics supplement?

  23. In most references, you do mention the DOI number, but in at least three references 48, 51, and 53 it is missing. This is inconsistent with the rest of the list.

Reviewer 4 Report

 Examining Adolescent Tennis Participation in Contemporary  China Using an Ecological Framework

This paper aims to provide a better understanding about the self-identified reasons for adolescents’ participation in non-organized or spontaneous tennis practice. Twenty-six adolescents and informants were recruited an participated in semi-structured interviews to provide descriptions and understanding of their continued tennis participation behaviors. Data were coded and analyzed via NVivo, from where four themes emerged: (i) Individual characteristics and self-interpretations of tennis culture; (ii) Microsystem mediating adolescents’ tennis participation; (iii) Barriers and obstacles toward tennis participation; (iv) Policies and microenvironments. The analysis led to conclusions that adolescent tennis participation is a result of the integration effect of the sociocultural and ecological factors.

 This study is an exploratory one which uses a qualitative analysis of 26 semi-structured interviews in order to find the main themes that underlie the issue in question. The interpretations of the finding were supported with an ecological framework, which includes elements from peoples’   microenvironments.

The paper is very well written, and all parts are presented in a complete manner, starting from the introduction section, which informs the reader adequately for the theme to be examine.

The procedures and the data analysis are presented in a clear way and with the details needed. It follows, in all steps of the qualitative analysis, a systematic application, which gives the paper credits. Even though it is just an exploratory work with no generalizable findings, in my opinion is an exemplar application of a qualitative methodology, and so I am positive for its publication.   

This ecological framework, that is fostered for describing and further for interpreting the emerged themes, is well organized and utilized for answering the research questions.

The limitations of the study were taken into consideration in the discussion and the concluding remarks. Since there are not many issues to reconsider in this fine piece of work, I recommend publication in this form.
